# Fatigue Evaluation for Innovative Excavator Arms Made of Composite Material

**DOI:** 10.3390/ma15217480

**Published:** 2022-10-25

**Authors:** Luigi Solazzi, Andrea Buffoli, Federico Ceresoli

**Affiliations:** Department of Mechanical and Industrial Engineering, University of Brescia (IT), Via Branze 238, 25123 Brescia, Italy

**Keywords:** structural lightening, composite materials, fatigue phenomenon in composite materials, numerical finite element analysis

## Abstract

This research reports the results related to the evaluation of the fatigue phenomenon of the arms of a medium–large excavator made of composite material (carbon fiber) instead of the classic constructional steel S355 (UNI EN 10025-3). In the numerical sizing phase, it was obtained that the overall weight of the excavator’s arms made of composite material is about 35% of the same components made of steel, obviously with equal performance in terms of the safety static coefficient, rigidity, and critical buckling load. The evaluation of the fatigue behaviour (assuming 5.25 × 10^6^ load cycles) applied for each load condition analyzed (levelling from the maximum distance to the minimum, lifting at the maximum distance, lifting at the minimum distance and rotation) shows the magnitude of the safety coefficients both related to the allowable stress and relative to the number of cycles acceptable. The assumption instead of combined cycles (involving one or more load conditions) leads to a significant reduction in the magnitude of the safety coefficients. The implementation of a loading cycle plan resulting from the different load conditions must be reliably assessed to evaluate as accurately as possible the fatigue behavior of the excavator arms made of composite material.

## 1. Introduction

Nowadays, the protection of the environment is of fundamental importance. The new solutions, for structural components, must be eco-friendly, eco-sustainable, and energy-efficient. The earthmoving machine sector is of paramount importance to achieve this target. A way to increase the efficiency of these machines is trying to reduce the CO_2_ consumption and to achieve this, it is also possible to operate on the weight of the components of these machines using innovative and performing materials. Examples of these materials can be aluminium alloys, composite materials, or sandwich, which are used to design the structural component in the automotive sector [1,2,3].

The use of composite materials also for the structural components [4,5] is very advantageous for many reasons; two of these are that they have very good mechanical properties and low density. Thanks to these properties, in recent years the possibility of applying composite materials in areas not strictly related to the automotive sector has been explored [6,7,8,9]. The added benefit involving the use of these materials is related to the inertia and accelerations reduction, which are very important factors in highly dynamic applications [10,11]. Research [12] has demonstrated that there is an advantage in terms of the system that commands components of earthmoving machines; in particular, cheaper and more compact systems are needed. In general, these components are preliminarily designed considering only static load conditions and assuming adequate safety factors (around three times with respect to the allowable strength of the material) [5,6,9,10]; however, during the real conditions works, a lot of components suffer the fatigue phenomena which involves their premature failure.

The aim of this research is to verify the fatigue resistance of an innovative structural component made of composite material; in particular, the handling arms, boom and stick, installed on an excavator. These components are subjected to continuous loading and unloading phases, phenomena that cause cyclic stress in time. In particular, it will be demonstrated how the composite materials respond to these types of loads compared to traditional materials commonly used. The study will be addressed using a composite made of epoxy resin and carbon fibers. Starting from some studies already addressed, in terms of static loads, on the excavator arms, the fatigue phenomena verification will be performed using numerical methods: Finite Element Method (FEM analysis).

The research was developed without considering the residual stresses present in the components, which can also be high and can be determined through the hole drilling method, digital image correlation (DIC) techniques, and numerically through FEM analysis [13]. This choice derives from the fact that the research also has an exploratory effect and therefore these important aspects will be further explored later.

## 2. Machine Description and Used Materials

The machine adopted, to conduct the optimization study relating to the lightening of the components, is a commercial excavator made by the Komatsu company (Tokyo, Japan) and more specifically the PC600 model (the year 2007). The choice fell on this machine because, in addition to being used for large uses, it is the same one already adopted for other studies and therefore has certain reliability in terms of design data.

The excavator has a total weight of 584,600 [N] and is characterized by a power of 323 [kW]; the geometric configuration is given by the presence of two arms, both marked in the figure: the curved boom with a length of 7.66 [m] and the stick which has a length of 3.5 [m]. Figure 1 shows the main excavator parameters and its working range.

The choice of the composite material [14,15] to be used for this application fell on the composition of epoxy resin and carbon fiber. In particular, a percentage of fiber of 60% and a matrix of 40% was chosen. Table 1 shows the main characteristics of the two phases:

To fully define the composite material, it is necessary to know the properties of the lamina along with the directions of the fiber. They are shown in Table 2 [14,15].

## 3. Load Condition

As shown in the load diagram in Figure 1, the excavator can assume an infinite number of geometric configurations for the two arms. Various mathematical modelling exists in the literature to describe the position, speed, and forces of the excavator’s arms [16,17,18,19].

In the present research, the arms were modelled in a complete manner, i.e., with all the parts that make up the arm system (arms, hydraulic cylinders, connecting rods, pins, etc.). This schematisation made it possible to avoid the development of numerical modelling of the excavator’s arms.

Four load conditions were assumed, representative of the heaviest load conditions to which the excavator may be subjected during its use [5].

### 3.1. Load Condition 1: Levelling from Maximum Distance to Minimum Working Distance

In this phase of work, the double articulation starts from a configuration where the hydraulic cylinder that controls the stick is in the fully retracted position, the bucket is therefore forced to perform a horizontal translation movement to level the ground (Figure 2).

### 3.2. Load Condition 2: Lifting with Full Load at the Minimum Working Distance

In this configuration, the excavator is lifting with a loaded bucket, placed at the minimum distance from the rotation axis, and is ready to be lifted vertically so that there are no variations in the angular position of the bucket to avoid the escape of the material. This phase, therefore, requires the hydraulic lifting cylinders, those that move the boom, and a considerable thrust in the extension phase, while the remaining cylinders are moved relatively to balance the position of the bucket (Figure 3).

### 3.3. Load Condition 3: Lifting Fully Loaded at the Maximum Working Distance

This load condition is similar to the previous one, however, the bucket is at the maximum distance from the rotation axis of the excavator. During movement, the bucket remains horizontal to prevent the exit of the material contained therein (Figure 4).

### 3.4. Load Condition 4: Rotation

This load condition occurs both in the levelling phase of the ground in the transverse direction, and in the event of material movement, therefore it is repeated many times on site. This load condition is particularly severe for the excavator arms, which are subjected to bending and torsional actions which are also a function of the position of the bucket with respect to the rotation axis of the machine (Figure 5).

## 4. Geometry of Arms Made of Composite Material

With the use of composite materials [20,21,22,23], from a manufacturing point of view and in regard to the stress state of the component, the creation of rectangular sections is somewhat problematic; for this reason, it was decided to opt for a different geometry and, more specifically, an elliptical section with a truncated cone with a variable thickness along its development is chosen. The truncated cone elliptical section was chosen for different reasons. First of all, as the arms work mainly in the vertical plane, the vertical plane is requested for greater stiffness. The second reason is correlated with the manufacturing process. Using the filament winding [24,25,26,27], it is possible to adopt the truncated conical core on which the composite material is wrapped, and once the process is finished, to remove the core and use it to make other arms.

In this way, it is possible to create a uniformly resistant section about the bending movements acting on the two planes. In particular, since the thickness of the composite is constant on the contour of the entire section, we proceeded by modifying the axes of the elliptical section to obtain a uniform resistance, that is, as can be seen from Figure 6, the stress at point A (*σ*_*zA*_) was imposed equal to the one at point B (*σ*_*zB*_) (Equation (1)). Regarding the bending movements on the two planes (*M_X_* and *d*
*M_Y_*), and to the properties of the section (*J_X_*, *J_Y_*, *W_X_*, *W_Y_*), Equations (1)–(5) reports the correlation between the movements acting on the two planes and the dimensions (*H*, *B*) of the elliptical section.
(1)$$\sigma_{ZA}=\sigma_{ZB}=\frac{M_{x}}{W_{x}}=\frac{M_{y}}{W_{y}}$$
(2)Jx=WxH2; Jy=WyB2
(3)b=B−2S; h=H−2S
(4)$$J_{x}=\frac{\pi }{64}B\times H^{3};\;J_{y}=\frac{\pi }{64}H\times B^{3}$$
(5)$$\frac{M_{x}}{M_{y}}=\frac{BH^{3}-\left(B-2S\right)\times {\left(H-2S\right)}^{3}}{B^{3}H-{\left(B-2S\right)}^{3}\times \left(H-2S\right)}\times \frac{B}{H}$$

Once the problem related to the basic geometry of the arm has been overcome, it is necessary to find a way to connect the various components necessary for its use, such as the hydraulic cylinders or the bucket.

To deal with this problem, it was decided to create special aluminum sections on which the various conical composite components will be grafted and glued.

This solution was designed to obtain continuity of the sections and in such a way as to not alter the resistance of the composite part by making holes for fixing as the latter are the trigger point, due to the stress concentration effect, for cracks (Figure 7).

Another characteristic of the composite material boom, as can be seen in Figure 8, is that it does not have the typical curvature that can be found in the two previous solutions. This choice was made for reasons of flexibility, modularity, and feasibility. In fact, by adopting composite sections with a truncated cone geometry with no curvature, the production process of the component is simpler and therefore less expensive; moreover, with this geometry, it is also easy to obtain different formats of the arms by replacing the different sections that constitute the arm itself. This geometry also allows us to obtain an excavation diagram not very different from the one obtainable from the classic solution, but what varies are the overall dimensions when the machine is stopped.

The theory used to size the arms was the Winding Angle [24,25].

Lastly, Table 3 shows the weight reduction [(Steel Weight—Comp. Weight)/Steel Weight)%] due to the use of composite material instead of classic steel.

## 5. Static Numerical Analysis for the Composite Arms

At this point, having defined the materials and the geometry, it is possible to proceed to the static numerical verification of the models of the arms, stick, and boom, presented in the previous paragraph.

The numerical analyses, carried out using the Autodesk Simulation Mechanical^®^ simulation software, are static and were conducted both for the initial phase and for the final phase of the movement according to the specific load condition.

The FEM model was made using solid brick elements with the quadratic formulation. These elements are solid elements with cubic or tetrahedron geometry. The characteristic is that these element types have nodes both at the vertices of and in the centre of the side forming the elementary solid. In general, the boom arm was designed with 35,000 elements, whereas the stick arm was designed with 20,000 elements.

For the FEM analysis of the composite arm, the simulation was performed on load condition 2, lifting at full load at the minimum working distance, with this being the most burdensome.

Figure 8 and Figure 9 show the results obtained in terms of stress and displacement for the work conditions 1,2, and 3.

Figure 8 and Figure 9 show that the component designed with innovative materials resists the stresses due to the heaviest operating situation.

For completeness, Table 4 shows the values of the maximum displacement for all the working conditions in which these values are in agreement with the one determined when the arms are built with classical structural steel.

## 6. Verify Fatigue Phenomena

Once defined and the geometry of the component minimizes its weight, it is possible to verify it to fatigue phenomena [28,29,30]. In this paragraph, these verifications will be addressed both through FEM analysis of the model defined above using Solidworks Simulation^®^ software (2020, Dassault Systèmes SolidWorks Corporation, Massachusetts, United States) and by analytical formulations.

### 6.1. Fatigue Loads Definition

The excavator arms that are the object of this study, during the working phase, are subjected to continuous load variations. These load variations involve stresses which are not constant and therefore induce fatigue stresses. The number of cycles to which the boom and stick are subjected is 525,000. This value derives from the hypothesis that the excavator works for 175 days per year, completing 200 cycles per day for 15 years at full load. The number of cycles was estimated from interviews with the operators.

The stress condition for each phase of the work was analysed for both boom and stick, in particular at the start and at the end of each one.

From the numerical phase, for the two components, we obtained the value shown in Table 5. The stress values, according to the Von Mises criterion, refer to the most stressed point both for the boom and stick.

The studied case is two, and they entail different ratios between the minimum and the maximum stresses. In the first case, fatigue verification has been performed using the maximum stresses at the start and at the end of each load condition, as if every single operation were repeated n times.

In this case, the ratio is equal to R = 0.1 because the minimum stress is not equal to 0 as the arms are always subject to the contribution of their own weight [6].

In the second case, instead, the stress condition is studied during the change in the load conditions, and, in particular, during the passage between the end of the levelling to the start of the lifting at the minimum distance. This can be considered a complete load condition.

This condition was addressed only for the stick being that this component is the most stressed. Table 6 shows the stress magnitude for two different load conditions at the start and finish movement.

### 6.2. A S-N Curve Definition

To deal with the calculation, it is necessary to know the S-N curve of the composite material and import it into the calculation software. The σ−N values regarding the stress ratio refer to the composite material made of epoxy resin matrix (40%) and carbon fiber (60%). Additionally, in this case, it is necessary to relate the number of cycles and the value of alternate stresses for both cycles’ ratios. To do this, it is possible to use the curves present in the literature [28,29,30]. These values have been processed in Excel ^®^ and imported into Solidworks^®^ to carry out the simulation.

Table 7 reports the values of alternate stress for R = 0.1 and R = 0.3 and shows the trend of the stress in the function of the number of cycles for both ratios. Where R is the ratio value between the minimum peak stress divided by the maximum peak stress. (R = σmin/σmax).

At this point, it is possible to proceed with the analytical analysis to verify the composite material fatigue resistance.

### 6.3. Analytical Approach to Fatigue Verification

This paragraph will address the analytical approach to verify that the excavator’s arms resist to the fatigue phenomenon. It is based on generic curves σ−N [30,31,32].

This verification will be applied to both cases, remember here:fatigue verification using the maximum stresses at the start and at the end of each works phase, for both boom and stick with R = 0.1;fatigue verification during the passage between the end of the levelling and the start of the lifting at the minimum distance, only for the stick, with R = 0.3.

Using the tabular values shown in Table 7 and the trend of the stresses in a cycle, the verification was carried out following what is reported in Table 6.

The value of the stresses for these cases is reported in Table 6 in the previous paragraph.

This verification, which is applicable to both composite and non-composite materials, is based on the determination of the safety factor expressed in terms of the number of cycles and stresses. Starting from a generalized Whöler diagram, it is possible to describe the course through a line in a bi-logarithmic diagram. The relation that describes this straight line is:(6)$$\sigma \times N^{k}=C$$

Knowing the value of fatigue stresses, in two points of the straight line, given the number of cycles, it is possible to derive the constants *k e C*. Moving in a bi-logarithmic system, the following is obtained:(7)logσ=−klogN+logC

Taking as the study points the beginning and end of the lines reported in graph 1, and using the values in the tables for R = 0.1 and R = 0.3, we obtained the values shown in Table 8. In this case, the number of reference cycles is from 10^3^ to 10^6^.

Replacement is obtained via the constant *k* and *C* which are reported in Table 9.

Now, it is possible to calculate the number of cycles to which the components resist when stressed at forces equal to those calculated by the numerical method.

The equation that permits calculation of the number of cycles is:(8)$$N={\left(\frac{C}{\sigma_{N}}\right)}^{\frac{1}{k}}$$

Additionally, their values for both conditions are reported in the Table 10.

Additionally, for the combinate cycle, end levelling, and start lifting at a minimum distance, the value in terms of the maximum number of cycles is reported in Table 11.

The component dimensioned for $525\times 10^{3}$ cycles can withstand the working conditions imposed. Now we evaluate the safety factors that result from these conditions.

Using the stresses at the $525\times 10^{3}$ listed in Table 5 and comparing the material performance, it is possible to evaluate the safety factor with respect to the stresses and number of the cycles using the following equations. The results are reported in Table 12 and Table 13.
(9)$$\eta_{\sigma }=\frac{\sigma_{t}}{\sigma_{w}}\;$$
(10)$$\eta_{N}={\left(\eta_{\sigma }\right)}^{\frac{1}{k}}\;$$

The obtained results demonstrate that the studied component can resist the fatigue phenomenon under the conditions analysed. The composite material dimensioned arms, therefore, in addition to ensuring a significant mass reduction, are able to face the real working phase, that is those that involve a variation of stress in time.

## 7. Conclusions

The aim of this research is the implementation of composite materials and, in particular, carbon fiber in the main components of an excavator (arms). The research focuses on evaluating the fatigue behaviour of the arms themselves. The analytical sizing and then the numerical verification by finite element analysis took place considering different load conditions to which the excavator may be subjected during its use. The implementation of composite materials (and aluminium alloy for the end parts) in place of the classic construction steel has made it possible to achieve a significant reduction in the overall weight of the arms. The total final weight is approximately 35% of the original weight. The sizing was carried out considering different aspects such as a static safety coefficient of approximately 3, a deformation or displacement magnitude comparable to the ones made of steel, and similar values in regards to the safety coefficient for the buckling phenomenon.

The peculiar aspect of this research lies precisely in the evaluation of the fatigue behaviour of the arms made of composite material. Based on interviews with users, a likely number of load cycles is approximately 5.25 × $10^{5}$. The excavator is used in a marble quarry on a non-continuous basis. A usage of 15 years × 250 days per year and ×140 cycles per day was assumed in agreement with the users. By adopting this value as an estimate, a series of safety factors were obtained both concerning stress and the number of cycles for each load condition analysed. In particular, the minimum safety coefficient with regard to the stress is equal to three and occurs for the load condition defined as the rotation. With the combination of several actions (assuming a cycle that consists of levelling the ground and lifting the bucket), the safety factor is drastically reduced. The assumption of a combined cycle or one involving several actions induces a significant reduction in the fatigue safety coefficient with regard to the arms made of composite material.

The development of the research concerns two aspects: the first is the experimental evaluation of the real use of a similar excavator made of steel; the second point relates to the evaluation in the operation field of the composite material with regards to the aspects related to fatigue and the influence of environmental conditions.

## Figures and Tables

**Figure 1 materials-15-07480-f001:**
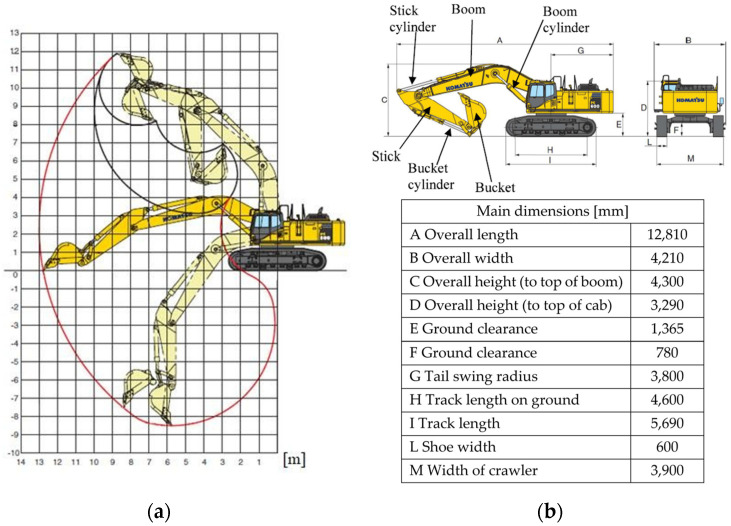
Excavator object of the study: (**a**) load diagram; (**b**) main dimensions.

**Figure 2 materials-15-07480-f002:**
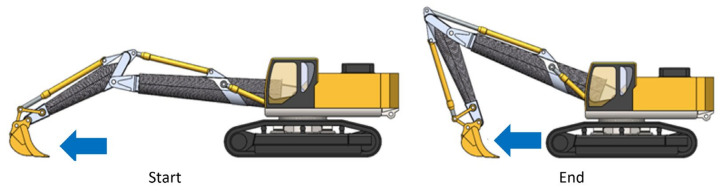
Load condition 1.

**Figure 3 materials-15-07480-f003:**
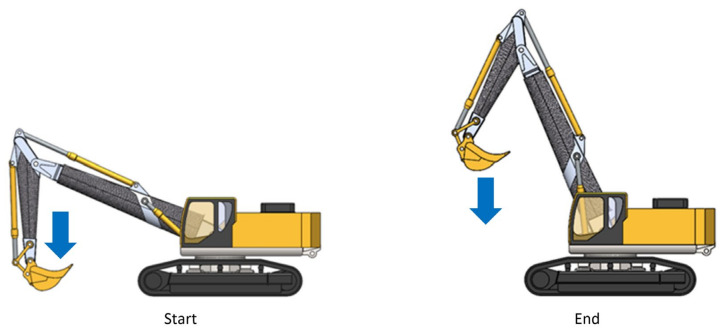
Load condition 2.

**Figure 4 materials-15-07480-f004:**
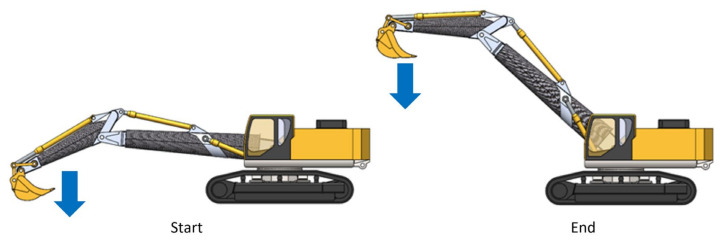
Load condition 3.

**Figure 5 materials-15-07480-f005:**
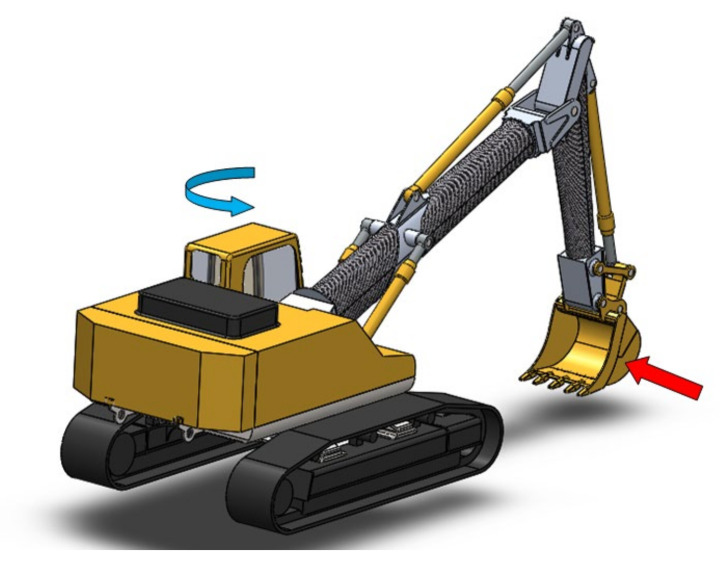
Load condition 4.

**Figure 6 materials-15-07480-f006:**
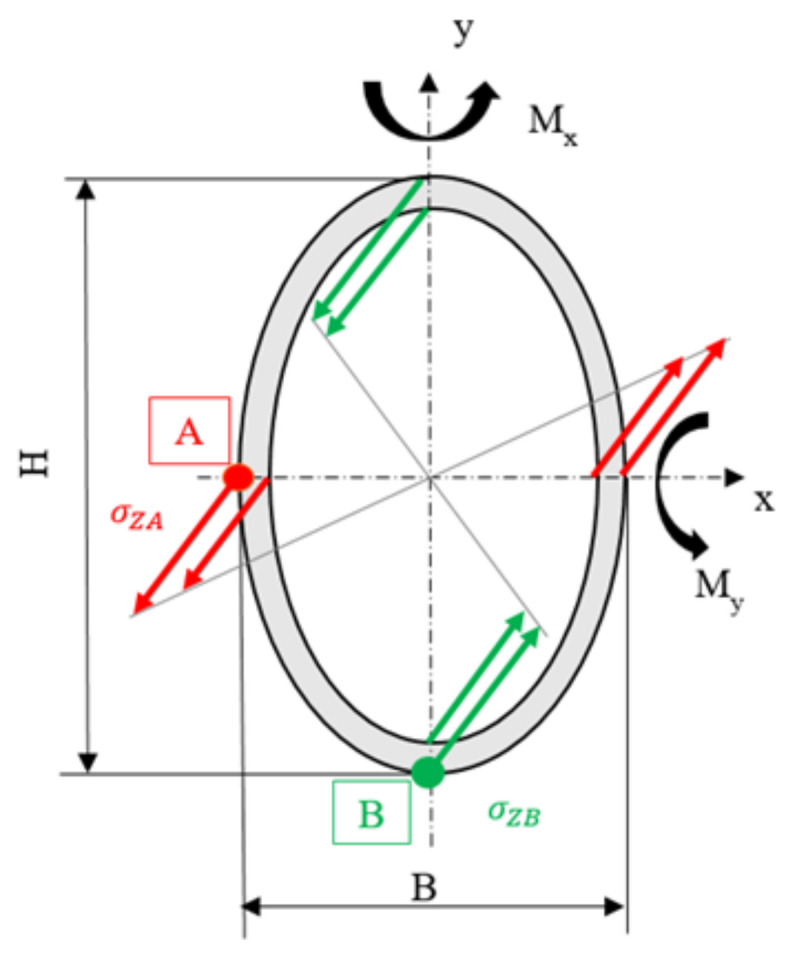
Elliptical section and its stress for the composite material.

**Figure 7 materials-15-07480-f007:**
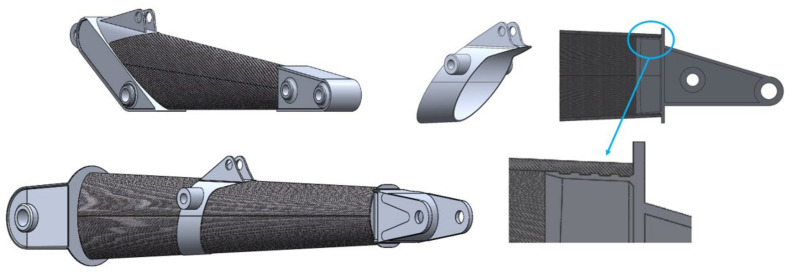
Geometrical solution for both the arms. The composite material is shown in dark grey; whereas, central and terminal elements made in the aluminium alloy are shown in light grey.

**Figure 8 materials-15-07480-f008:**
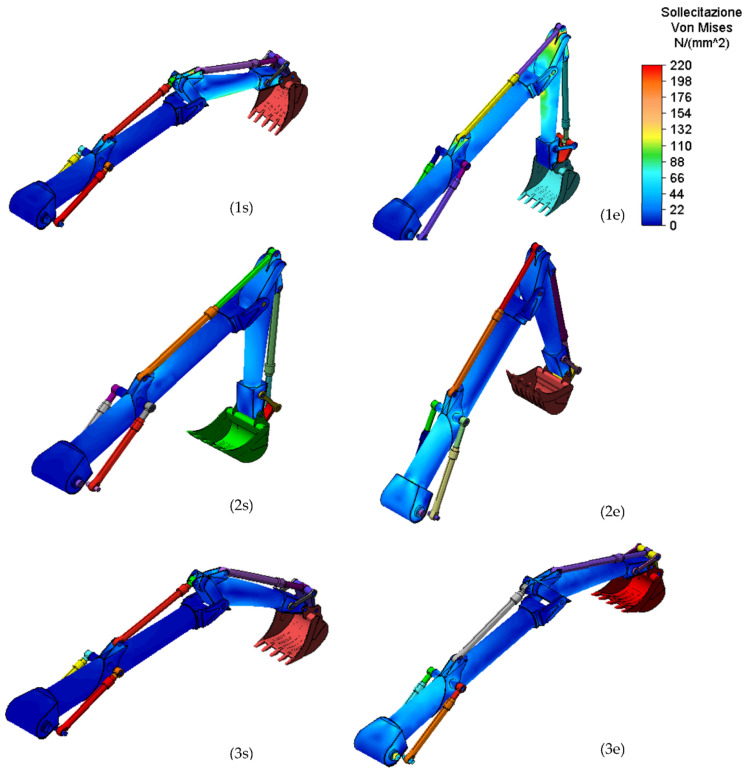
Maximum Von Mises Stress, for stick and boom, for the work conditions: start (**s**) and end (**e**) movement for the 1, 2, and 3 load conditions.

**Figure 9 materials-15-07480-f009:**
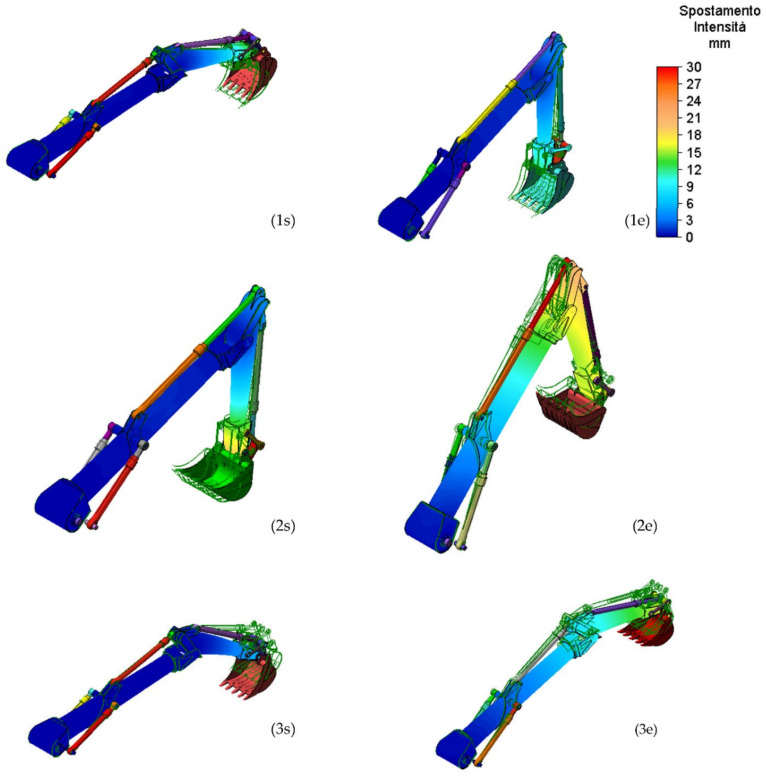
Displacements for the work conditions 1, 2, and 3 for both boom and stick: start (**s**) and end (**e**) movement.

**Table 1 materials-15-07480-t001:** Mechanical properties of composite material.

Material	Density ρ [kg/m^3^]	Young’s Modulus E [MPa]	Shear Modulus G [MPa]	Poisson Ratio
Epoxy resin	1200	4500	1600	0.4
Carbon fiber	1750	230,000	50,000	0.3

**Table 2 materials-15-07480-t002:** Mechanical properties of composite material in direction of the lamina.

Properties	Density ρ [kg/m^3^]	Young’s Modulus E1 (|| to the Fibres) [MPa]	Young’s Modulus E2 (⊥ to the Fibres) [MPa]	Shear Modulus G12 [MPa]	Major Poisson’s Ratio ν12	Minor Poisson’s Ratio ν21
Composite	1530	139,800	10,929	3817	0.34	0.026

**Table 3 materials-15-07480-t003:** Weight reduction using composite material.

Material	Weight Boom [N]	Weight Reduction [%]	Stick Weight [N]	Weight Reduction [%]	Total Weight [N]	Weight Reduction [%]
S355 (UNI EN 10025-3)	36,000	-	15,000	-	51,000	-
Composite	12,450	65.4	5850	61.0	18,300	64.1

**Table 4 materials-15-07480-t004:** Maximum displacements [mm] for the steel and composite arms.

Material	Load Conditions					
	1		2		3	
	Start	Finish	Start	Finish	Start	Finish
S355 (UNI EN 10025-3)	7.3	10.1	9.8	15.1	5.6	6.2
Composite	7.0	9.2	18.1	16.7	8.5	10.5

**Table 5 materials-15-07480-t005:** Start and end stresses [MPa] for the stick and boom.

	Stick		Boom	
	Start	Finish	Start	Finish
Levelling	103.8	122.6	8.9	51.8
Lifting max distance	33.2	27.2	3.9	25
Lifting minimum distance	40.7	19.8	9.8	49.9
Rotation	157.5	32.7	6.3	6.9

**Table 6 materials-15-07480-t006:** Start and end stresses [MPa] for the stick during the phase change.

	Stick	
	Start	Finish
Levelling	103.8	122.6
Lifting minimum distance	40.7	19.8

**Table 7 materials-15-07480-t007:** Cycles–Alternate Stress with R = 0.1 for composite material chosen.

Composite Material	R = 0.1	R = 0.3
Cycles	Alternate Stress [MPa]	Alternate Stress [MPa]
10	780	270
100	770	255
1000	720	241
10,000	650	220
100,000	580	193
1,000,000	530	160

**Table 8 materials-15-07480-t008:** Number of cycles and their relative stresses for R = 0.1 and R = 0.3.

	$\sigma_{N}\;[\mathrm{MPa}]$	
$N_{cycles}$	R = 0.1	R = 0.3
10^3^	720	241
10^6^	530	160

**Table 9 materials-15-07480-t009:** Number of cycles and their relative stresses for R = 0.1 and R = 0.3.

Ratio	*k*	*C* [MPa]
R = 0.1	0.044	880
R = 0.3	0.044	297

**Table 10 materials-15-07480-t010:** Number of cycles at the maximum stress for R = 0.1.

R = 0.1		Stick	Boom
Start		N° cycles at the maximum stress [MPa]	N° cycles at the maximum stress [MPa]
	Levelling	$1.25\times 10^{21}$	$2.20\times 10^{45}$
	Lifting at max. distance	$2.23\times 10^{23}$	$3.06\times 10^{53}$
	Lifting at min. distance	$2.17\times 10^{30}$	$2.46\times 10^{44}$
	Rotation	$9.59\times 10^{16}$	$5.66\times 10^{48}$
End			
	Levelling	$2.84\times 10^{19}$	$9.07\times 10^{27}$
	Lifting at max. distance	$2.07\times 10^{34}$	$1.40\times 10^{35}$
	Lifting at min. distance	$2.8\times 10^{37}$	$2.12\times 10^{28}$
	Rotation	$3.15\times 10^{32}$	$7.16\times 10^{47}$

**Table 11 materials-15-07480-t011:** Number of cycles at the maximum stress for the combinate cycles (R = 0.3) for the stick arm.

Combinate cycle: end levelling and Start Lifting minimum distance	$4.74\times 10^{8}$

**Table 12 materials-15-07480-t012:** Safety coefficient for the stick arm in the case of R = 0.1.

Start		$\eta_{\sigma }$	$\eta_{N}$
	Levelling	4.6	$1.44\times 10^{15}$
	Lifting at max. distance	14.5	$2.57\times 10^{26}$
	Lifting at min. distance	11.8	$2.50\times 10^{24}$
	Rotation	3.0	$1.10\times 10^{11}$
End			
	Levelling	3.9	$3.27\times 10^{13}$
	Lifting at max. distance	17.7	$2.38\times 10^{28}$
	Lifting at min. distance	24.3	$3.24\times 10^{31}$
	Rotation	14.7	$3.62\times 10^{26}$

**Table 13 materials-15-07480-t013:** Safety coefficient for the stick arm in case of the combinate cycle (R = 0.3).

	$\eta_{\sigma }$	$\eta_{N}$
Combinate cycle: End Levelling and Start Lifting minimum distance	1.45	$4.63\times 10^{3}$

## Data Availability

The data presented in this study are available on request from the corresponding author.

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
