# Peer review of "Fatigue Evaluation for Innovative Excavator Arms Made of Composite Material"

_materials, 2022, doi:10.3390/ma15217480_

Round 1

Reviewer 1 Report

1. The author needs to check the manuscript for the English grammar and spelling such as “plausible” and “thabout” used in line no. 320 and 324 respectively.

2. The literature review part is very weak and author need to improve it by including different approaches of mathematical modeling of excavator arm.

3. Author need to explain that why in table 4 the variation in displacement of  composite and steel arm is nearly 2 times of for 2 load condition. 

4. Author have not explained that “R” stands for what, at place of its first use.

5. Author have taken elliptical section in stress analysis without proper justification. Author needs to give valid reason to take elliptical section. 

6. Author have done the fatigue analysis at 5.25x105 stress cycle based on the interview of the operator but not clarified how much is the experience of operator/operators and how many number of operator author have interviewed. It will be better to take standard data rather than taking such a critical input based on interview. 

7. There are many references those have not been cited in the manuscript such as [14]  [16]  to [20] etc.

Author Response

Dear Reviewer, thank ypu very muach your obeservations. I attach the response file.

Thank you  again.

Reviewer 2 Report

I have finished my review comments in the attachment file. In general, it is an interesting article in terms of the static performance of composite materials. 

However, there are some shortcomings in terms of English expressions, static test verification, S-N curve accuracy and improvement of analysis software. 

Author Response

Dear Reviewer, thank you very much for your observations. I attach the response file.

Thank you again.

Reviewer 3 Report

Kindly consider the attached file

Author Response

(The authors gave the same response as above.)

Round 2

Reviewer 2 Report

Maybe my comments were not considered by you.